# Influence of the Grain Size Distribution of the Limestone Additives on the Color Properties and Phase Composition of Sintered Ceramic Materials Based on Cream-Firing Clays

**DOI:** 10.3390/ma15072694

**Published:** 2022-04-06

**Authors:** Kornelia Wiśniewska, Ewelina Kłosek-Wawrzyn, Radosław Lach, Waldemar Pichór

**Affiliations:** Department of Building Materials Technology, Faculty of Materials Science and Ceramics, AGH University of Science and Technology, al. Mickiewicza 30, 30-059 Kraków, Poland; eklosek@agh.edu.pl (E.K.-W.); radoslaw.lach@agh.edu.pl (R.L.); pichor@agh.edu.pl (W.P.)

**Keywords:** color, phase composition, cream-firing clays, limestone, sintering process

## Abstract

The study focused on determining color changes in materials made of cream-firing clays from the Opoczno region (Poland) due to the addition of calcium carbonate in the form of limestone. Moreover, the influence of the grain size distribution of this additive on the color properties of the materials and their phase composition was determined. Test samples were prepared using theplastic method and fired at four different temperatures: 1120, 1140, 1160 and 1180 °C. The color properties of the surface of ceramic materials were determined in CIE-Lab color space using a colorimeter. Quantitative phase analysis was performed using the Rietveld method. The research showed that the addition of calcium carbonate causes an increase in the yellow color factor and a decrease in the red color factor and the brightness of the material. Moreover, it was proven that the grain size distribution of the additive used significantly influences the phase composition of the materials, thus determining the values of physical properties and the color of the materials.

## 1. Introduction

It is well known that the grain size distribution of raw additives used in ceramic masses has a significant impact on the properties of ceramic materials. It was proven that granulation of additives influences the porosity of the final material, its phase composition and microstructure and thus its technological properties [1,2]. Moreover, it has been noticed that different grain size distributions of dolomite flour (one of the additives used in ceramic industry) cause changes in the color of materials fired under the same conditions [1]. Dolomite is frequently used as a flux in the ceramic industry. Flux agents are widely used in order to modify the range of the firing temperature [3]. It is well known that the firing temperature has a significant impact on the technological properties of ceramic products, such as compressive strength, porosity, water absorption [4,5] and frost resistance [6]. One of the raw materials that functions as a flux (besides dolomite) is calcium carbonate, which is added to mass as a firing shrinkage reducer. The most common form of calcium carbonate used in the industry is the widely available limestone. This additive affects the final properties of ceramic products not only due to the reduction in porosity caused by the action of the flux, but also due to its chemical composition, which affects the color properties of obtained materials and their phase composition. In the case of the addition of calcium carbonate, the grain size distribution is a crucial factor that influences the properties of ceramic materials due to the harmfulness of calcium carbonate with a grain diameter above 0.5 mm [7].

The adverse impact of coarse carbonates on the properties of ceramic materials is widely known. The decarbonation of carbonate grains with diameters over 0.5 mm leads to the formation of coarse grains of calcium oxide. These grains react with water vapor to form calcium hydroxide. Reducing the density from 3.34 g/cm^3^ to 2.21 g/cm^3^ during the reaction leads to cracking and the destruction of the fired material [7].

However, calcium carbonates with a grain diameter smaller than 0.5 mm are not classified as harmful additives. Their addition to ceramic masses affects the high levels of porosity and water absorption of ceramic materials as a result of the decarbonation process [7]. Ceramic building materials obtained from calcareous clays at low temperatures close to the onset of clay sintering are characterized by lower mechanical strength in comparison to those obtained from non-calcareous raw materials [8]. Moreover, these materials are damaged (microcracked) in the course of frost resistance testing. This phenomenon is no longer evident once the products are fired at higher temperatures [8], because formed CaO acts as a flux. Based on data in the subject literature, it cannot be clearly determined whether fine-grained carbonate additions have a beneficial or adverse effect on the properties of ceramic materials.

There have been several studies concerning the effect of calcium carbonate on the color of ceramic materials made of red and yellow clays [9,10,11]. When red clay materials are fired at low temperatures with the addition of calcium carbonate, bright efflorescence may form on the surface of the resulting ceramics, leading to a significant increase in the brightness of the sintered products [8]. Another mechanism of color change is calcium-iron-aluminosilicates formation. The output color depends on the CaO/FeO molar ratio and the degree of CaO and FeO conversion to new phases [10,11]. According to Seger, the light or dark red color of the iron-containing clays turns to yellow or gray-yellow if 3–3.5% wt. of CaCO_3_ is added to the mass for every 1% wt. of Fe_2_O_3_. The more the CaCO_3_ content exceeds the specified minimum, the brighter the yellow color [10]. Sandfort examined the CaO-Al_2_O_3_-Fe_2_O_3_ system and found that the red color of Fe_2_O_3_ is better maintained and more resistant to the effects of firing when it forms a solution with Al_2_O_3_. A yellow color is obtained when only small amounts of Fe_2_O_3_ are present in the solution; with larger amounts, a red color is obtained. A yellow color is also formed in mixtures containing Fe_2_O_3_ with relatively large amounts of CaO and Al_2_O_3_ [11].

In this study, a series of tests were performed in order to understand the impact of the addition of limestone (characterized by different grain size distributions) on the color of sintered ceramic materials. These materials could be used in the manufacture of clinker tiles. Clinker tiles are an attractive building material not only because of their constructional and technological values, but also their visual qualities [12]. The aim of this study was to obtain sintered ceramic materials without the need to cover them with glaze. Therefore, it was necessary to obtain a material with a homogeneous color of the surface. Moreover, since these materials are intended to be used in the manufacture of unglazed products, it has been an important issue to determine how the addition of limestone shapes the color of a material’s surface. A tendency to change the brightness and color (shades of blues and yellow) of the surface of the materials was also determined in accordance with the grain size distribution of the limestone. Such relationships have already been determined and extensively described for the addition of dolomite with different grain size distributions [1,2] and using different sintering temperatures [2].

## 2. Materials and Methods

This study dealt with the technological and color properties of ceramic materials based on Borkowice clay as the main raw material and limestone from Lhoist Bukowa Sp.z.o.o. (Bukowa, Poland) as the color-changing additive. Chemical analysis of used raw materials was performed with theXRF method (WDXRF AxiosmAX spectrometer, PANalytical, Malvern Panalytical, Almelo, The Netherlands) and is presented in Table 1.

Borkowice clay belongs to clay resources of the Opoczno region (Poland) and it is characterized by lightcolor-firing properties [13] and a low content of coloring oxides, according to Table 1. The literature describes the nature of the Borkowice clay as kaolinite–illite raw material [13]. The aforementioned properties of this raw material allow for its wide use in the ceramic tile production industry [13]. Moreover, such characteristics of the clay raw material make it possible to verify the influence of various types of mineral raw materials on the color of the ceramic materials produced during the firing. L,a,b color indicators for Borkowice clay have been determined (L: 85.35, a: 1.23, b: 7.79) [1]. The phase composition of Borkowice clay has been determined, and it indicates the presence of large amounts of kaolinite, illite and quartz [1,2]. The semiquantitative analysis of the kaolinite:illite:quartz ratio is 5:4:1 [2]. The mineralogical composition has a strong effect on the behavior of fired samples; thus, that is of decisive importance for the quality and properties of the final materials [14].The analysis of the Borkowice clay grain size distribution was performed using the hydrometer analysis for clays and soils [15], and it is shown in Table 2.

Analysis of the particle size distribution of Borkowice clay showed that the greatest share of grains is characterized by a size smaller than 1 μm, because of a high content of clay minerals, such as kaolinite and illite. This raw material was defined as fine-grained. Jurassic limestone from the Bukowa deposit was used as a color-changing additive in ceramic masses. Bukowa limestone is considered to be one of the purest varieties of Polish limestone due to high content of CaO (54.93%) and low content of admixtures, according to Table 1. The phase composition of limestone from the Bukowa deposit is characterized by a calcite content of 99%. The remaining phase is quartz [16]. Tests of limestone from the Bukowa deposit showed that it is characterized by a high value of total porosity (16.1%) compared to limestone from other geological periods: 1.4% for the Devonian limestone from the Trzuskawica deposit and 0.7% for the Lower Carboniferous limestone from the Czatkowice deposit. Moreover, limestone from the Bukowa deposit is characterized by the presence of finer calcite crystals compared to limestone from the deposits in Trzuskawica and Czatkowice [16]. The listed properties, such as the size of the crystallites and porosity of the limestone, play significant roles in increasing the rate of the calcination and thermal decomposition of limestones [17] during the firing process. The experiment assumed the addition of limestone from the Bukowa deposit in four different grain sizes in order to determine the influence of this raw material grain size on the color and technological properties of final sintered ceramic materials. Four types of limestone were prepared depending on their grain size via grinding in an agate mill: B4, B15, B105 and B316. The analysis of grain size distribution of all types of limestone was characterized using laser diffraction analysis (Malvern Mastersizer2000 analyzer, Malvern Panalytical, Almelo, The Netherlands), and results of this analysis are shown in Figure 1.

The prepared raw material was milled in order to obtain a large spread of the dominant grain size, from 4.37 μm for the finer limestone variety to 316 μm for the coarsest limestone variety.

In order to conduct the research, 5 types of ceramic masses were prepared, and their compositionsare presented in Table 3.

The tested masses and sintered samples were prepared in accordance with laboratory conditions. The first step was to grind the limestone to appropriate grain sizes (Figure 1) using an agate mill. Then, the dry components of the masses (according to the composition provided in Table 3) were mixed with the addition of water to impact plasticity. The masses prepared in the described way were stored in humid conditions for 24 h. After 24 h of storage, laboratory unglazed tiles in two different dimensions were formed: 150 × 30 × 20 mm (for flexural strength) and 75 × 30 × 20 mm (for other analyses). Samples were formed via extrusion in a laboratory vacuum extruder with auger. After being formed, samples were dried in humid conditions for 24 h. Samples were positioned at a suitable distance to allow air circulation. The next step was to dry the samples in a laboratory dryer to the constant mass with the following process steps: 35 °C for the first 6 h, then the temperature was increased to 50 °C, 75 °C and 105 °C every 2 h. The last stage of drying was maintaining the temperature of 105 °C for 7 h. The firing process was performed in a laboratory electric kiln in an oxidizing atmosphere with firing rate: 100 °C/h and dwelling time: 1 h at selected temperatures: 100 °C, 600 °C and 900 °C and 2 h at maximum temperature. Samples were fired at four different temperatures: T1 = 1120 °C, T2 = 1140 °C, T3 = 1160 °C and T4= 1180°C.

After sample preparation, the technological properties, phase composition, microstructure and color of the sintered materials formed during firing were tested. The following materials properties were determined: total water absorption by boiling in water and by 24 h of soaking in the water [18,19], open porosity [18], total shrinkage and flexural strength [20] in accordance with European standards. The microstructure analyses were carried out using the SEM electron microscope Nano Nova SEM (FEI, Hillsboro, OR, USA) and presented in the images with ×500 magnification for selected samples. The phase compositions of the sintered ceramics were determined via XRD analyses (X’Pert Pro, Phillips, Malvern Panalytical, Almelo, The Netherlands) with the following equipment parameters: radiation source: copper anode; angle range (2θ): 5–90°; angle step: 0.008°, using a reference intensity ratio (RIR) method [21], with the 10% addition of zinc as the reference material. The quantitative analysis of crystalline phases and amorphous phase was determined with the Rietveld method [21] using the X’PertHighScore Plus program (PANalytical). The color of the surface of the sintered ceramic materials was determined using Colorimeter 3Color CP-10 (3Color^®^, Jodłowa Poland) using CIELab color space with a CIE 10°/D65 observer. Three factors characterize the CIELab color space: L—luminance (brightness), a—color from green to red and b—color from blue to yellow. Consequently, the CIELab color space is described as a 3D coordinate system in which L indicator values extend from 0 to 100, a indicator values—from −150 to +100 and b indicator values—from −100 to +150. Green and blue shades are determined by negative (−) numerical values of parameters a and b, respectively, while red and yellow shades are assigned as positive (+) values of parameters a and b, respectively [22].

## 3. Results and Discussion

### 3.1. Physical Properties

The results of technological properties of sintered ceramics (total water absorption, total open porosity, total shrinkage and flexural strength) are summarized and shown in Figure 2. The measurements were made in four different samples in order to determine the error. The properties of ceramic materials change with the increase in the firing temperature and the change in the grain size of the limestone additive.

According to Figure 2a,b, a decrease in open porosity and water absorption for materials with the addition of the limestone, compared to the reference material, was observed. This phenomenon can be explained by the fact that larger-diameter limestone grains do not undergo complete transformation during the firing process. The exception is the material with the addition of the finest graining of limestone MB4 (dominant grain size: 4.37 μm), for which values of total open porosity and total water absorption are higher than for the reference material. This tendency was expected due to the transformation of the fine-grained calcium carbonate during the firing process, which led to an increase in the open porosity and thus water absorption of the ceramic materials [23,24]. For materials fired at 1120 °C, 1140 °C and 1160 °C, there was a significant decrease in the discussed values between materials with the addition of the finest type of the limestone and the remaining materials. Then, as the dominant grain size of the additive increased, the values of porosity and water absorption did not change significantly. However, for the coarsest graining of the additive tested, an increase in these values was observed. For materials fired at the highest temperature—1180 °C—no increase in these values was observed for the coarsest type of the limestone. Moreover, there was a strong dependence between total water absorption (and thus, total open porosity) and the firing temperature. Total water absorption and total open porosity decreased with increasing firing temperature, which was caused by the sintering process [25]. According to Figure 2c, it was observed that the addition of limestone caused a decrease in linear shrinkage during the thermal treatment of the samples and that the linear shrinkage increased with the increase in dominant grain size of the used additive. Total shrinkage did not change significantly with increasing the temperature. The highest shrinkage was recorded for the M0 material, and the lowest for the MB4 material. The flexural strength of all obtained ceramic materials was greater than 19 MPa. The described phenomenon is the result of two mechanisms opposing each other: the intensification of contraction by the formation of a liquid phase and the reduction in contraction by reactions. As a result of the supersaturation of the alloy with calcium ions, the crystallization of the calcium-aluminum-silicate phases began. Crystallization from anorthite and wollastonite alloy was manifested by a reduction in the linear sintering shrinkage [26]. Small limestone grains reacted faster to the amorphous phase and anorthite than the large ones. As shown in Figure 2d, it was found that the addition of limestone caused a decrease in the flexural strength value regardless of the firing temperature, which was expected according to the previous research [27]. The higher the firing temperature, the greater the difference between the flexural strength values of the M0 material and of other materials (with the addition of limestone). It was noticed that an increase in the grain size of the applied limestone additive increased the flexural strength of the material (except for MB316 material). This phenomenon can be also explained by the fact that larger-diameter limestone grains do not undergo complete transformation during the firing process and therefore act as matrix enhancers. In the case of materials with the addition of limestone, the temperature is not a factor determining the value of the mechanical strength of the material.

### 3.2. Color Properties

In order to determine the visual utility value of the materials, the color parameters in the CIELab color space were determined. The measurement was made at three different locations on the sample. The results obtained are shown in Figure 3. In addition, photos of the 2.5 × 2.5 cm cut from the surfaces of the materials are shown in Figure 4.

Based on the data contained in Figure 3, an increase in the b indicator, a decrease in the a indicator and a decrease in the L indicator with increasing temperature was noticed. The exception was the MB4 material, for which the values of a and L indicators remained relatively constant, regardless of the firing temperature. It was noticed that regardless of the firing temperature, materials with limestone addition (regardless of grain size) were characterized by a higher value of the b indicator, a lower value of the a indicator and a lower value of the L indicator than the reference material M0. The following exceptions were distinguished: MB306 had a slightly lower value of the b indicator, a higher value of the a indicator and a higher value of the L indicator than the reference material: M0 at lower firing temperatures. Moreover, the MB4 material was characterized by higher values of a and L indicators than the reference material, regardless of the firing temperature. When analyzing the surface color of the materials with the naked eye (according to Figure 4), it could be noticed that the surface of MB306 material seemed to be much brighter than the surface of other materials. It was not, however, caused by directing the color indicators towards a white color, but by the appearance of hydrated residues after the decarbonation of limestone on the surface of the products. The appearance of these grains is an unfavorable phenomenon, leading to the destruction of products. This phenomenon occurs in materials with the addition of coarse-grained limestone fractions [7]. Lime grains were observed on the surface of MB306 materials fired at all temperatures and MB105 fired at lower temperatures: 1140 °C and 1160 °C.

### 3.3. Phase Composition Analysis

The dependency between the amount of crystalline components and the amount of amorphous phase and the size of CaCO_3_ grains was only observed for samples fired at 1160 and 1180 °C. The results are shown in Figure 5. The phase composition of KB306 was omitted in the analysis due to the lime grains formed on the surface of the ceramic material.

The phase composition analysis of sintered materials with the determination of the amount of amorphous phase shows that all materials contained the basic crystalline phases that characterize sintered ceramics materials: mullite (Al_4+2x_Si_2−2x_O_10−x_) and quartz (SiO_2_).Mullite is formed during the kaolinite decomposition reaction at a temperature of about 900–1000 °C. Quartz is formed during high-temperature firing processes of silicate materials (containing, for example, kaolinite) [28]. In some fired materials, the presence of a small amount of rutile (TiO_2_) was also observed, which occurred as a consequence of the chemical composition of the clay raw material (Borkowice clay), in which it occurs as an impurity (Table 1). The high-temperature transformations that occurred during the firing process also led to the formation of a large amount of amorphous phase. After the addition of limestone to the ceramic masses, anorthite (Ca[Al_2_Si_2_O_8_]) appeared in the phase composition of the materials, the presence of which depended on the chemical composition of the added limestone (Table 1). Apart from the anorthite, the phase composition of MB105 contained gehlenite (Ca_2_Al[SiAlO_7_]). Gehlenite appeared at the expense of a reduced amount of anorthite in the phase composition with an increase in the grain size of the calcium carbonate additive used. The phase composition of MB306 contained a small amount of anorthite due to the anorthite crystallization model [26]. However, there was a relatively higher presence of another crystalline phase: portlandite (Ca(OH)_2_), the presence of which was responsible for the appearance of lime grains on the surface of the fired materials (according to Figure 4).

On the basis of the data contained in Figure 5, it was observed that for the materials fired at 1160 °C, there was a significant increase in the amorphous phase content in the phase composition of the materials with the increase in the dominant grain size of the limestone additive used. On the other hand, for materials fired at 1180 °C, there was a decrease in the amorphous phase content in the phase composition with an increase in the dominant grain size of the limestone. Both for materials fired at 1160 °C and 1180 °C, the amount of amorphous phase was greater for materials with the addition of the limestone than for the reference materials. For materials fired at 1160 °C, there was a decrease in anorthite content in the phase composition of materials with an increase in the dominant grain size of the limestone additive used, while for materials fired at 1180 °C—an increase in anorthite content in the phase composition of materials with an increase in the dominant grain size of the limestone. Quartz and mullite contents remained constant regardless of the grain size of the limestone. Therefore, it can be concluded that the grain size distribution of the added limestone influences the share of the amorphous phase and anorthite in the phase composition of the materials. Moreover, it was noticed that the anorthite and the amorphous phase have opposite relationships to each other. As the share of the amorphous phase in the phase composition increased, the share of anorthite decreased. This relationship proves the ongoing crystallization of the anorthite from the amorphous phase.

### 3.4. Microstructure Analysis

The microstructure analysis of selected ceramic materials (M0, MB4, MB15 and MB105) fired at 1160 °C is presented in Figure 6, Figure 7, Figure 8 and Figure 9. For the presentation of the results, a magnification of 500× was used to show changes in the microstructure depending on the grain size distribution of the calcium carbonate additive.

By analyzing the image of the microstructure of the reference material M0 and the results of the EDS determination, it could be seen that the microstructure included kaolinite clusters and rutile grains scattered in an aluminosilicate matrix.

Figure 7, Figure 8 and Figure 9 show images of the microstructure of materials with the addition of limestone of different grain size. The measurement of pores formed as a result of calcium carbonate decomposition shows that the larger the grain size of the limestone introduced into the mass, the larger the pores formed after the firing process. However, the microstructure images show that the large limestone grains did not completely decompose in comparison to the small limestone grains. Figure 10 shows the microstructure analysis of the MB306 material fired at 1160 °C at 500× and 3000× magnification.

A large amount of lime waste accumulated around the large pores, which was also confirmed via the EDS analysis and XRD analysis (Figure 5)—the appearance of gehlenite and portlandite in the phase composition of MB105 and MB306 materials, respectively. The accumulation of the lime waste around the large pores led to the appearance of lime grains on the surface of the fired materials, which led to the cracking and destruction of the sintered material. Figure 10b shows the microstructure of MB3016 material with ×3000 magnification. In this figure, an accumulation of limestone residue placed around large pores formed as a result of calcium carbonate reactions during the firing process can be seen. Limestone residue was also scattered in the aluminosilicate matrix, causing a change in its chemical composition. The same CaCO_3_ dissolution mechanism was observed for all samples containing the addition of limestone, fired at any temperature. The greater the grain size distribution of the used limestone, the more lime residue is dissolved in the aluminosilicate matrix.

## 4. Conclusions

The conducted research allowed us to find the relationship between the grain size distribution of the added limestone and the color of the surface of the final product and its phase composition. In addition, these tests allowed for the selection of the limestone graining interval, the use of which would enable a homogeneous color of the product surface. The following conclusions were made during the experiment:As the firing temperature increases, the value of b indicator increases, and the values of a and L indicators decrease. This means that as the sintering process progresses, the materials based on cream-firing clays are characterized by higher contents of yellow and red shades in their color and by lower brightness.The addition of calcium carbonate causes an increase in the value of the b indicator and a decrease in the values of a and L indicators. Therefore, the introduction of limestone to mass based on cream-firing clays results in materials with a higher intensity of yellow and red shades and lower brightness being obtained.The addition of the finest-grained calcium carbonate causes an increase in the value of the L color indicator of the obtained materials. Therefore, the use of fine-grained limestone as an additive to cream-firing clays allows one to obtain a material withhigh brightness.Large limestone grains do not completely decompose during the sintering process. This results in the accumulation of lime waste around large grains and the appearance of portlandite in the phase composition of these materials. This leads to the appearance of lime grains on the surface of the materials, which give the impression of lightening the surface but also lead to cracking and the destruction of the product.The addition of the finest variety of limestone increases the total open porosity and therefore the total water absorption of sintered material. The remaining types of calcium carbonate reduce the total open porosity and therefore the total water absorption of sintered material.The addition of calcium carbonate causes an increase in total shrinkage values. The bigger the dominant grain size of used limestone, the higher value of total shrinkage.The addition of calcium carbonate causes a decrease in the flexural strength values.With the addition of limestone to the ceramic mass, a new phase appears in the phase composition of the material: anorthite.The grain size distribution of the limestone addition does not influence the share of quartz and mullite in the phase composition of materials.The grain size distribution of the limestone addition influences the share of amorphous phase and anorthite in the phase composition of the materials.The amount of amorphous phase is greater for materials with the addition of the limestone than for the reference material.As the share of the amorphous phase in the phase composition increases, the share of anorthite decreases. This is proof of the ongoing crystallization of the anorthite from the amorphous phase.

## Figures and Tables

**Figure 1 materials-15-02694-f001:**
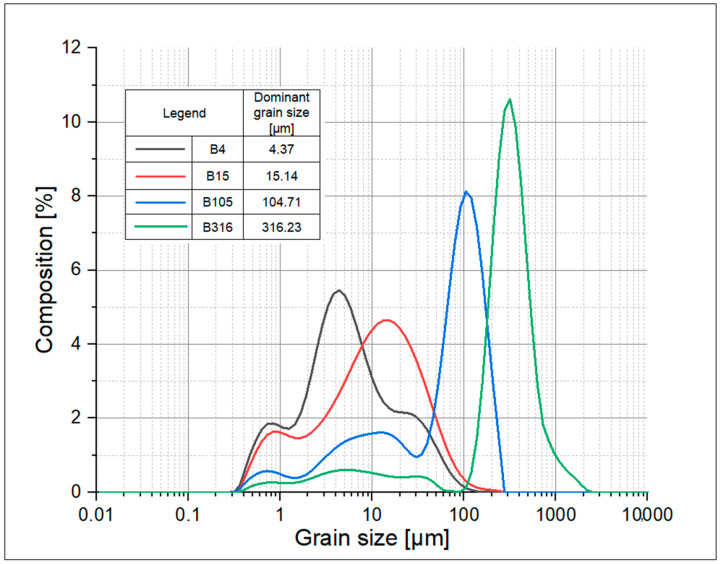
Grain size distribution of limestone from Bukowa deposit in four different grain sizes: B4, B15, B105 and B316.

**Figure 2 materials-15-02694-f002:**
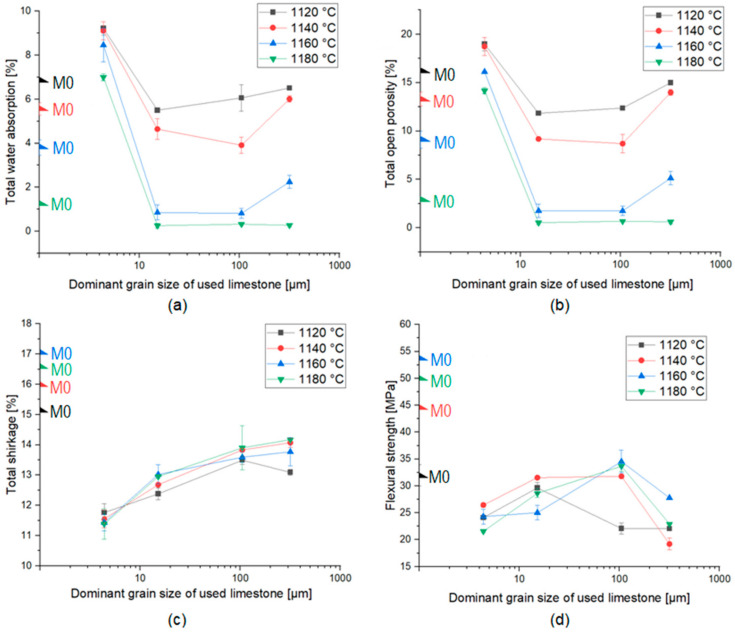
Results of technology properties analysis of fired ceramic materials: (**a**) total water absorption, (**b**) total open porosity, (**c**) total shrinkage, (**d**) flexural strength.

**Figure 3 materials-15-02694-f003:**
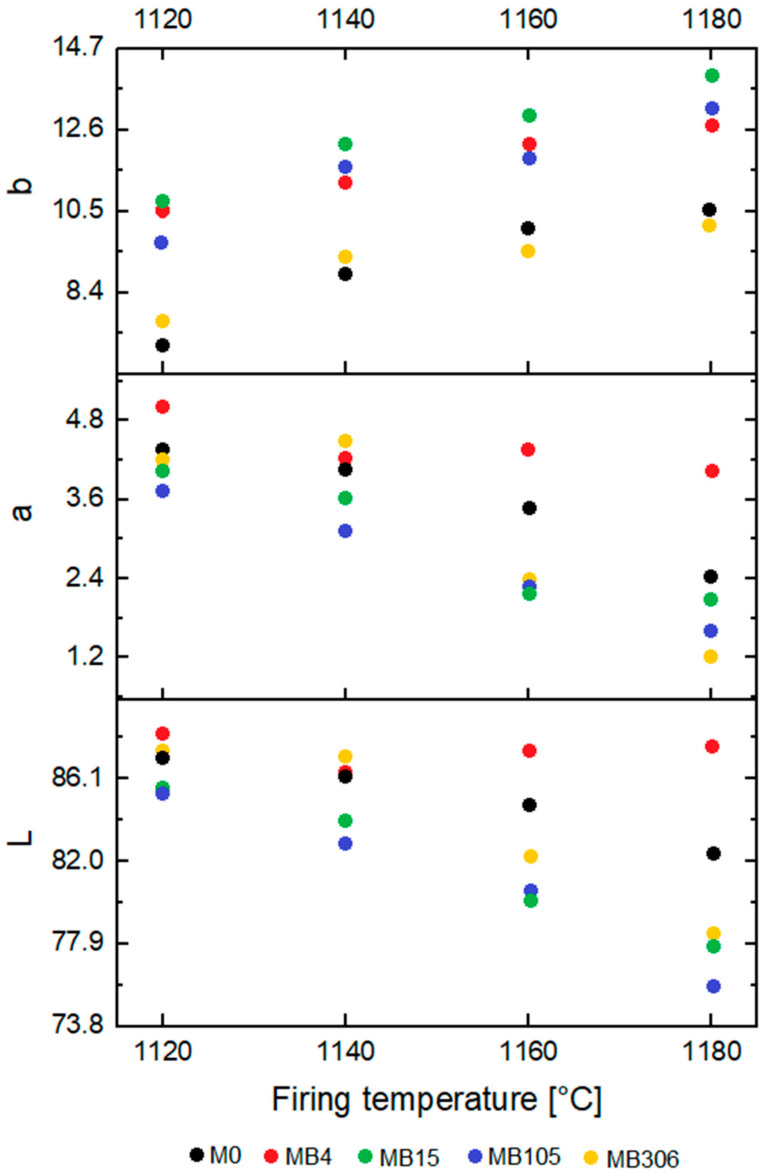
Color analysis of obtained ceramic materials: L, a and b color indicators for materials fired at: 1120, 1140, 1160 and 1180 °C.

**Figure 4 materials-15-02694-f004:**
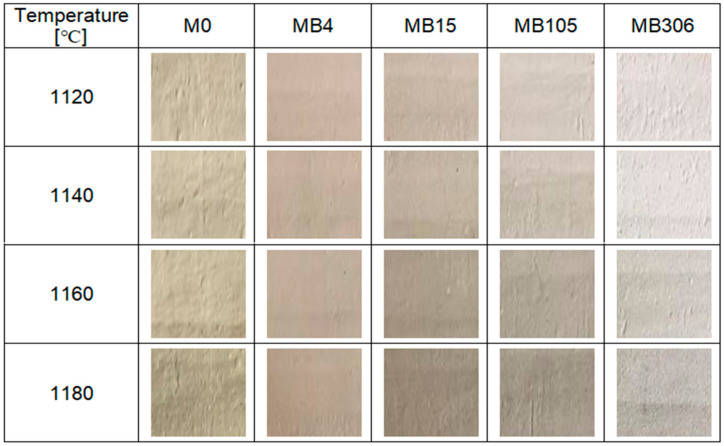
Color of the surface of obtained ceramic materials.

**Figure 5 materials-15-02694-f005:**
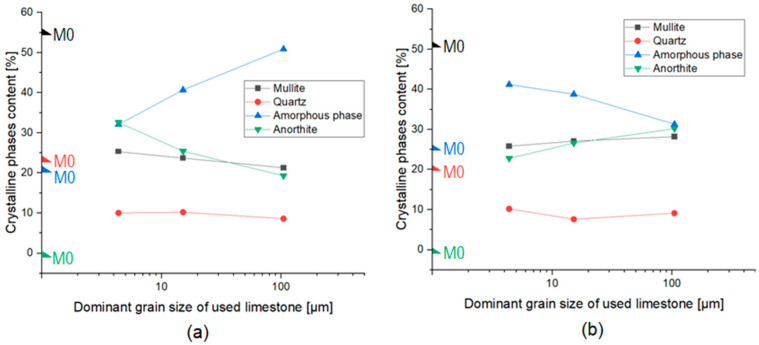
Phase composition analysis of materials fired at: (**a**) 1160 °C and (**b**) 1180 °C.

**Figure 6 materials-15-02694-f006:**
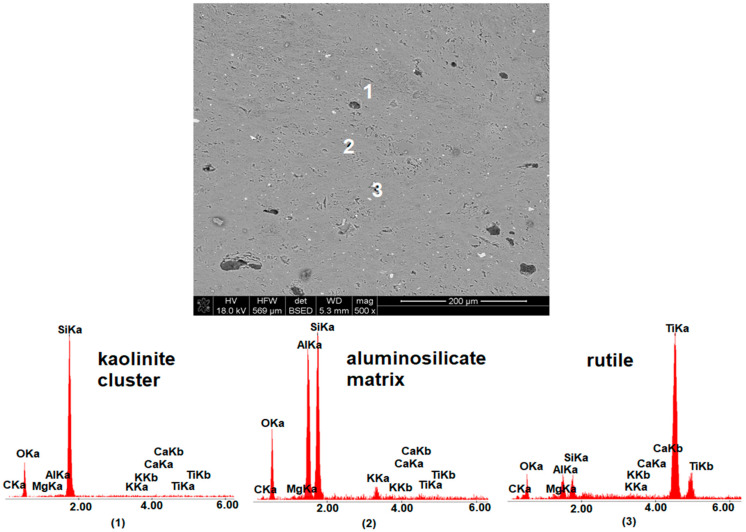
SEM-EDS image of M0 material fired at 1160 °C.

**Figure 7 materials-15-02694-f007:**
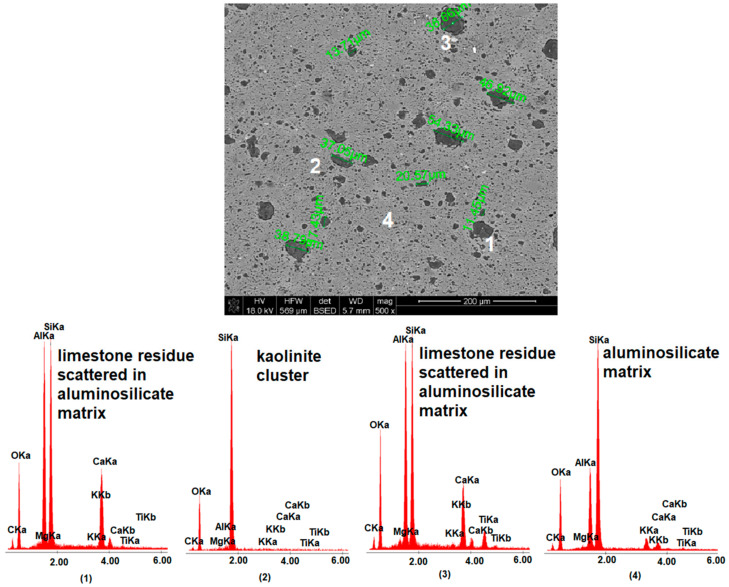
SEM-EDS image of MB4 material fired at 1160 °C, including measurement of the size of pores resulting from the decomposition of calcium carbonate.

**Figure 8 materials-15-02694-f008:**
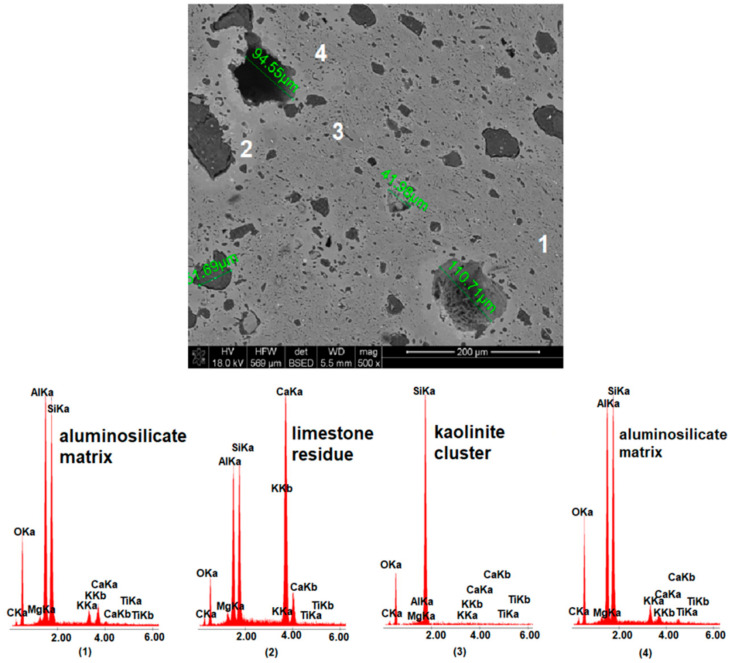
SEM-EDS image of MB15 material fired at 1160 °C, including measurement of the size of pores resulting from the decomposition of calcium carbonate.

**Figure 9 materials-15-02694-f009:**
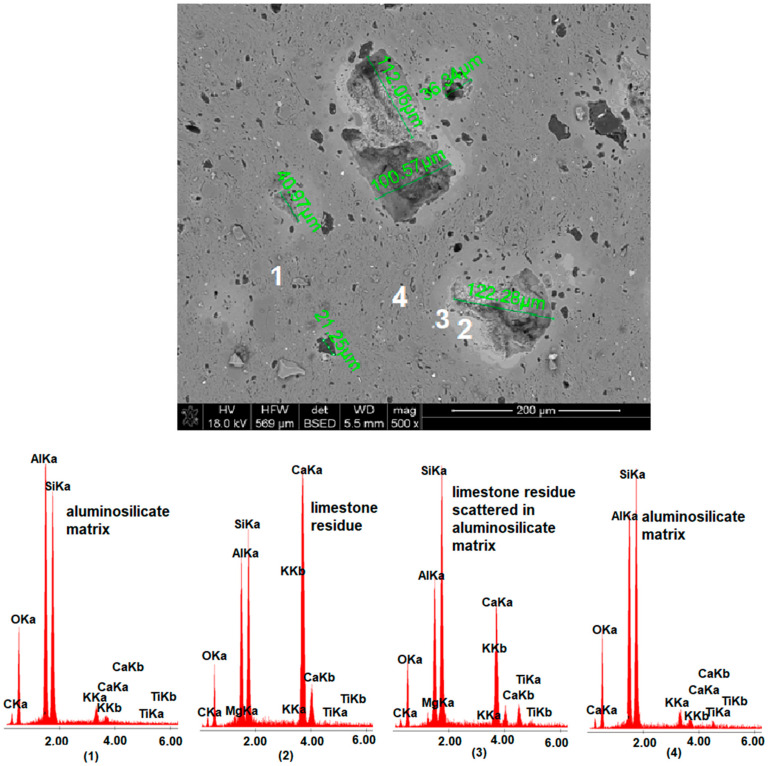
SEM-EDS image of MB105 material fired at1160 °C, including measurement of the size of pores resulting from the decomposition of calcium carbonate.

**Figure 10 materials-15-02694-f010:**
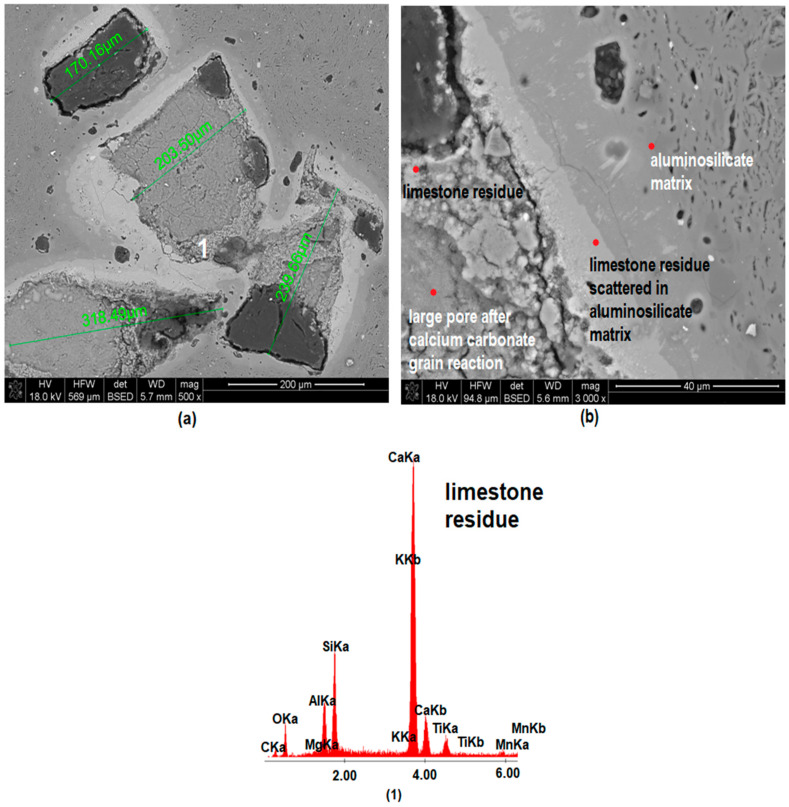
SEM-EDS image of MB306 material fired at 1160 °C, including measurement of the size of pores resulting from the decomposition of calcium carbonate with (**a**) ×500 magnification and (**b**) ×3000 magnification.

**Table 1 materials-15-02694-t001:** Chemical analysis of raw materials: Borkowice clay and Bukowa limestone.

Raw Material	Chemical Composition (wt/%)												
	SiO_2_	Al_2_O_3_	Fe_2_O_3_	TiO_2_	CaO	MgO	K_2_O	Na_2_O	Cr_2_O_3_	P_2_O_5_	MnO	SO_3_	Loss on Ignition (Organic Material)
Borkowice clay	53.22	30.92	0.99	1.34	0.23	0.41	1.60	0.16	0.02	0.15	0.00	0.02	10.73
Bukowa limestone	0.72	0.19	0.05	0.00	54.93	0.41	0.19	0.05	0.00	0.01	0.00	0.00	43.34

**Table 2 materials-15-02694-t002:** Particle size distribution of Borkowice clay.

**Grain Fraction (μm)**	>63	32–63	16–32	8–16	4–8	2–4	1–2	<1
**Frequency (%)**	0.1	5.9	2.9	8.8	10.1	11.5	11.7	49

**Table 3 materials-15-02694-t003:** Composition of ceramic masses.

Name of the Material	Amount of Raw Material (wt/%)				
	Borkowice Clay	Bukowa Limestone			
		B4	B15	B105	B316
M0	100	0	0	0	0
MB4	90	10	0	0	0
MB15	90	0	10	0	0
MB105	90	0	0	10	0
MB316	90	0	0	0	10

## Data Availability

The data are contained within the article. Additional data are available on request from the corresponding author.

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
