# Peer review of "Influence of the Grain Size Distribution of the Limestone Additives on the Color Properties and Phase Composition of Sintered Ceramic Materials Based on Cream-Firing Clays"

_materials, 2022, doi:10.3390/ma15072694_

Round 1
Reviewer 1 Report
(1) The English language of this manuscript must be improvrd.
(2) The discussion and conclusion shouldn't be mixed in one section.
Author Response
Dear Reviewer,
Thank You sincerely for the review. We are submitting our revised manuscript through the Author Center.
Below We are sending responses to Your comments.
The English language of this manuscript must be improved.
The manuscript has been re-edited for linguistic correctness.
The discussion and conclusion shouldn't be mixed in one section.
Conclusions are separated from discussion. The discussion is linked to the results.
We hope that the improvements we have made to the manuscript and to the attached figures are clear and sufficient.
We are looking forward to Your response.
Best regards,
Kornelia Wiśniewska
AGH University of Science and Technology, Kraków, Poland

Reviewer 2 Report
This paper presents a lot of results and looks of good quality at first glance. To obtain a paper of good quality, the results must be properly discussed. The literature survey is scarce. The evaluation criteria of the quality of the obtained products are not clear. The reason for producing the pale color of the products is not presented.
- The biggest problem is the insufficient use of literature. Every result that is obtained should be explained, the question "Why?" is to be on our minds for any phenomenon that has taken place, and researchers are obliged to try to answer that. Results and discussion are often convenient to be presented together, which includes the comparison of the obtained results to the literature data. The Conclusion must begin with an introductory sentence in which the basic idea of the study shown is presented.
- Each standard referred to by the authors should be in the list of references. The stated standard 14411 represents technical conditions. According to which 10545 standard was water absorption, porosity, flexural strength tested? Based on what are the results compared to determine the products as candidates for external use? All this must be written. Every claim in scientific writing must be clear. Besides, to determine whether the tiles are good for outdoor usage, it is obliged to check the resistance to frost, which was not done in this work. Please, bear in mind that the titles are, among other criteria, differed by the methodology of production as dry-pressed and extruded tiles. Both kinds have their parts of 14411 and 10545 standards.
- The end of the introduction should summarize what was done in the research, and why. What is the most significant of what has been discovered? Has anyone else done similar research, and what has been found? You can consider, as an example, this paper about ceramic tiles, which is good to lead your further thinking: https://doi.org/10.1016/j.bsecv.2020.11.006.
- Why were the papers published by a similar group of authors previously not cited in this work and the results here obtained compared? The subject there presented is similar to this one, only the addition of MgCO3 and its fractions is tried to increase the lightness. Why are the phases obtained by firing clay not compared with the mineral composition of the starting clay? Was this the same sample of clay examined and published previously or not? It is important to determine the mineralogical content of the initial clay used in the mix to be able to discuss which mineral transformations happened during the firing process. Did the initial clay contain quartz and in which quantity? Is there more or less of it after the firing?
- A scheme for the production of laboratory products is unnecessary, and a textual passage in this regard would take up less space. What are "beams", do you mean laboratory unglazed tiles?
Figures are generally hardly visible and the quality of the images must be improved. The text in SEM images is not clear.
SEM images with EDS analyses do not show much. The magnification used is too small to give useful results. Clay minerals are below 2 μm, and a huge area of 200 μm is examined. It is not clear what the points refer to, given that there are numerical marks on the two images, and the EDS results do show what they refer to. Based on which formulas the location of a kaolinitic matrix is determined. Is there still kaolinite in the fired materials, or is it mullite? How can kaolinite contain the most (and almost only) of silicon?
- The mentioned efflorescence for the sample of the palest color does not look like the mentioned phenomenon in the photograph. How was water absorption determined? Capillary absorption or a vacuum? If the absorption was capillary (by immersion in distilled water at room temperature), did Ca(OH)2scum appear in the water?
- The essence of the work is not clear. Why is this light color important and has been sought for? What is the practical meaning of it? For what type of ceramic products? Tiles are mentioned only later in the text, and masonry is also mentioned in one place. In addition, which mineralogical changes does the color change follow?
- It is not entirely true that CaCO3grain size below 0.5 mm is so important in itself. The composition of the raw clay to which it is added is also of very high significance. If the clay does not contain enough clay minerals, for example, diopside, which increases the mechanical strength, cannot be formed. You can refer to 10.2298/HEMIND121123006A for more information.
- Have you considered including a mentor with experience on the subject? It can be very useful for this work and your future studies. If you do not have someone available near you, read, ask, contact other scientists whose works you have read.
- In the introduction, during the analysis of the literature data, it is not enough to state the color of the studied clay. What is much more important is which minerals it contained and in which quantity.
- The Abstract is 19 % plagiarized, or the text is similar to other published papers. Please, rephrase.
- Please change “loss of ignition” to “loss on ignition” and state what is lost by the ignition.
- Replace “Grainfraction” with “Grain fraction”.
Author Response
Dear Reviewer,
Thank You sincerely for the review. We are submitting our revised manuscript through the Author Center.
Below We are sending responses to Your comments.
This paper presents a lot of results and looks of good quality at first glance. To obtain a paper of good quality, the results must be properly discussed. The literature survey is scarce. The evaluation criteria of the quality of the obtained products are not clear. The reason for producing the pale color of the products is not presented.
Thank You for this comment. The discussion is now linked to the results and expanded. Moreover we developed the literature survey by 12 items. The evaluation criteria of the quality of the obtained products and the reason for producing this products are presented in the Introduction:
“The aim of this study was to obtain sintered ceramic materials without the need to cover them with glaze. Therfore, it was necessary to obtain a material with a homogeus color of the surface. Moreover, since these materials are intended to be used in the manufacture of unglazed products, it has been an important issue to determine how the addition of limestone shapes the color of the material’s surface. A tendency to change the brightness and color (shades of bues and yellow) of the surface of the materials was also determined in accordance with the grain size distribution of the limestone.”
The biggest problem is the insufficient use of literature. Every result that is obtained should be explained, the question "Why?" is to be on our minds for any phenomenon that has taken place, and researchers are obliged to try to answer that. Results and discussion are often convenient to be presented together, which includes the comparison of the obtained results to the literature data. The Conclusion must begin with an introductory sentence in which the basic idea of the study shown is presented.
The literature has been developed by 12 items. Results are now explained properly (text marked in red in: Results and Discussion). The discussion is now linked to the results. The conclusion begins with the following text: “The conducted research allowed to find the relationship between the grain size distribution of the added limestone and the color of the surface of the final product and its phase composition. In addition, these tests allow for the selection of the limestone graining interval, the use of which would enable a homogeneous color of the product surface”.
Each standard referred to by the authors should be in the list of references. The stated standard 14411 represents technical conditions. According to which 10545 standard was water absorption, porosity, flexural strength tested? Based on what are the results compared to determine the products as candidates for external use? All this must be written. Every claim in scientific writing must be clear. Besides, to determine whether the tiles are good for outdoor usage, it is obliged to check the resistance to frost, which was not done in this work. Please, bear in mind that the titles are, among other criteria, differed by the methodology of production as dry-pressed and extruded tiles. Both kinds have their parts of 14411 and 10545 standards.
Thank you for this observation. 10545 standard has been added as [18] and [20] in References. Moreover ASTM C373-18:2018 has been addend as [19] in references and it refers to the boil method for extruded ceramic tiles. The criteria for the final product are described in the Introduction.
The end of the introduction should summarize what was done in the research, and why. What is the most significant of what has been discovered? Has anyone else done similar research, and what has been found? You can consider, as an example, this paper about ceramic tiles, which is good to lead your further thinking: https://doi.org/10.1016/j.bsecv.2020.11.006.
The introduction has been supplemented with missing data.
“In this study series of tests have been performed in order to understand the impact of the addition of the limestones (characterized by the different grain size distribution) on the color of sintered ceramic materials. These materials could be used in the manufacture of clinker tiles. Clinker tiles are an attractive building material not only because their constructional and technological values, but also visual qualities [12]. The aim of this study was to obtain sintered ceramic materials without the need to cover them with glaze. Therfore, it was necessary to obtain a material with a homogeneous color of the surface. Moreover, since these materials are intended to be used in the manufacture of unglazed products, it has been an important issue to determine how the addition of limestone shapes the color of the material’s surface. A tendency to change the brightness and color (shades of bues and yellow) of the surface of the materials was also determined in accordance with the grain size distribution of the limestone. Such relationships have already been determined and extensively described for the addition of dolomite of different grain size distribution [1,2] and using different sintering temperatures [2]”. Moreover, the proposed article was used as [25]. Thanks for the recommendation.
Why were the papers published by a similar group of authors previously not cited in this work and the results here obtained compared? The subject there presented is similar to this one, only the addition of MgCO3 and its fractions is tried to increase the lightness. Why are the phases obtained by firing clay not compared with the mineral composition of the starting clay? Was this the same sample of clay examined and published previously or not? It is important to determine the mineralogical content of the initial clay used in the mix to be able to discuss which mineral transformations happened during the firing process. Did the initial clay contain quartz and in which quantity? Is there more or less of it after the firing?
Thank you for this observation. We supplemented the citation with the indicated publication [2]. Mineral composition of the starting clay is also supplemented: “Phase composition of Borkowice clay has been determined and it indicates the presence of large amount of kaolinite, illite and quartz [1,2]. The semiquantitive analysis of kaolinite:illite:quartz ratio is 5:4:1[2]. Mineralogical composition has a strong effect on the behavior of fired samples; thus that is of decisive importance for the quality and properties of the final materials [14]” in 2. Materials and Methods.
A scheme for the production of laboratory products is unnecessary, and a textual passage in this regard would take up less space. What are "beams", do you mean laboratory unglazed tiles?
As You suggested, a scheme for the production of laboratory products was replaced with a textural passage:
“The first step was to grind the limestone to appropriate grain size (Figure 1) usiang an agate mill. Then, the dry componentsof the masses (according to the composition provided in Table 3) were mixed with the addition of water to impact plasticity. The masses prepared the described way were stored in humid conditions for 24 hours. After 24 hours of storage, laboratory unglazed tiles in two different dimensions were formed: 150x30x20 mm (for flexural strength) and 75x30x20 mm (for other analysis). Samples were formed by extruding in laboratory vacuum extruder with auger. After forming, samples were being dried in humid conditions for 24 hours. Samples were positioned at proper distance to allow air circulation. Next step was to dry the samples in laboratory dryer to the constant mass with following process steps: 35 ℃ for the first 6 hours, then the temperature was increased to 50 ℃, 75 ℃ and 105 ℃ every 2 hours. The last stage of drying was maintaining the temperature of 105 ℃ for 7 hours. Firing process was performed in laboratory electric kiln in oxidizing atmosphere with firing rate: 100 ℃/h and dwelling time: 1 hour at selected temperatures: 100 ℃, 600 ℃ and 900 ℃ and 2 hours at maximum temperature. Samples were fired at four different temperatures: T1= 1120 ℃, T2= 1140 ℃, T3= 1160 ℃, T4= 1180℃”.
The phrase "beams" has been replaced by the phrase “laboratory unglazed tiles”.
Figures are generally hardly visible and the quality of the images must be improved. The text in SEM images is not clear.
The quality of all figures has been improved and the text has been enlarged.
SEM images with EDS analyses do not show much. The magnification used is too small to give useful results. Clay minerals are below 2 μm, and a huge area of 200 μm is examined. It is not clear what the points refer to, given that there are numerical marks on the two images, and the EDS results do show what they refer to. Based on which formulas the location of a kaolinitic matrix is determined. Is there still kaolinite in the fired materials, or is it mullite? How can kaolinite contain the most (and almost only) of silicon?
A magnification of 500x was used to show changes in the microstructure depending on the grain size distribution of the calcium carbonate additive (measurement of the size of pores resulting from the decomposition of calcium carbonate). As You suggested a magnification of 3000x is presented for one of the materials in order to show the CaCO3 dissolution mechanism. The EDS references were improved by showing the microstructure for samples fired at one temperature only. The place of presence of the aluminosilicate matrix was determined according to the EDS analysis (aluminum, silicon). Fired ceramic materials typically contain mullite and quartz as shown in Figure 5.
The mentioned efflorescence for the sample of the palest color does not look like the mentioned phenomenon in the photograph. How was water absorption determined? Capillary absorption or a vacuum? If the absorption was capillary (by immersion in distilled water at room temperature), did Ca(OH)2scum appear in the water?
The resulting phenomenon was misnamed. The phrase "efflorescence" has been replaced by the phrase “grain”. Water absorption was determined by boiling in water and by 24 hours of soaking in the water which has been specified in the text and references to the standards have been added.
The essence of the work is not clear. Why is this light color important and has been sought for? What is the practical meaning of it? For what type of ceramic products? Tiles are mentioned only later in the text, and masonry is also mentioned in one place.
Thank You for this comment. The essence of the work has been explained it the Introduction:
“In this study series of tests have been performed in order to understand the impact of the addition of the limestones (characterized by the different grain size distribution) on the color of sintered ceramic materials. These materials could be used in the manufacture of clinker tiles. Clinker tiles are an attractive building material not only because their constructional and technological values, but also visual qualities [12]. The aim of this study was to obtain sintered ceramic materials without the need to cover them with glaze. Therfore, it was necessary to obtain a material with a homogeneous color of the surface. Moreover, since these materials are intended to be used in the manufacture of unglazed products, it has been an important issue to determine how the addition of limestone shapes the color of the material’s surface. A tendency to change the brightness and color (shades of bues and yellow) of the surface of the materials was also determined in accordance with the grain size distribution of the limestone. Such relationships have already been determined and extensively described for the addition of dolomite of different grain size distribution [1,2] and using different sintering temperatures [2]”.
In addition, which mineralogical changes does the color change follow?
Such tests have been performed, but in this case the influence of the phase composition on the color is negligible.
It is not entirely true that CaCO3grain size below 0.5 mm is so important in itself. The composition of the raw clay to which it is added is also of very high significance. If the clay does not contain enough clay minerals, for example, diopside, which increases the mechanical strength, cannot be formed. You can refer to 10.2298/HEMIND121123006A for more information.
Thank you for that point. The article you recommended turned out to be very helpful in considering the influence of mineralogical composition on the technological properties of products. It has been quoted as [14].
In the introduction, during the analysis of the literature data, it is not enough to state the color of the studied clay. What is much more important is which minerals it contained and in which quantity.
Color analysis for studied clay has been attached: “L*a*b color indicators for Borkowice clay has been determined (L: 85.35, a: 1.23, b: 7.79) [1]”.
Semiquantitive analysis of Borkowice clay has also been attached: “Phase composition of Borkowice clay has been determined and it indicates the presence of large amount of kaolinite, illite and quartz [1,2]. The semiquantitive analysis of kaolinite:illite:quartz ratio is 5:4:1[2]”.
The Abstract is 19 % plagiarized, or the text is similar to other published papers. Please, rephrase.
According to Materials Office similarity to a previously published work is slightly high at 8%. These similarities has been rephrased, according to Your suggestion.
Please change “loss of ignition” to “loss on ignition” and state what is lost by the ignition.
Thank You for noticing this error. It has been corrected.
Replace “Grainfraction” with “Grain fraction”.
Thank You for noticing this error. It has been corrected.
We would like to thank You for the useful tips that allowed us to improve our manuscript and we hope that our responses to the comments are transparent and sufficient.
We are looking forward to Your response.
Best regards,
Kornelia Wiśniewska
AGH University of Science and Technology, Kraków, Poland

Reviewer 3 Report
In this article, the authors study the impact of the granulometry of limestone incorporated at 10 wt% in one type of clay, then fired to produce sintered ceramics. 5 properties are characterized after firing: water absorption, open porosity, total shrinkage, flexural strength and colour.
First of all, some remarks on the form:
- Many figures are unreadable (figure 2, figure 3, figure 6, figure 8,9 and 10)
- The size of the zone on the photographs of the samples should be precised (figure 5)
Then, remarks on the methodology and the results:
- There’s a lack of discussion of the results: for example on the graph showing the flexural strength vs the dominant grain size of limestone for the four firing temperature (figure 3), we see the curve for 1120° is different than the others: this result is not explained.
- Line 220 – 224: the authors asses than « only the materials fired at 1160 and 1180° are suitable for external use »: but which are the results that allow to conclude that, and on which norm or known results it relies?
- On figure 3, errors bars are represented for some results but not for all the values (or maybe they’re so small that they can’t be seen?) and we haven’t any information about the number of tests carried on to calculate these error bars.
- For the colour measurements: how did the authors checked the homogeneity of the samples? The photographs show some differences
Author Response
Dear Reviewer,
Thank You sincerely for the review. We are submitting our revised manuscript through the Author Center.
Below We are sending responses to Your comments.
Many figures are unreadable (figure 2, figure 3, figure 6, figure 8,9 and 10)
The quality of all figures has been improved and the text has been enlarged.
The size of the zone on the photographs of the samples should be precised (figure 5)
Thank You for this comment. This information has been supplemented: “…of the 2.5 x 2.5 cm cut from the surfaces of the materials…”
There’s a lack of discussion of the results: for example on the graph showing the flexural strength vs the dominant grain size of limestone for the four firing temperature (figure 3), we see the curve for 1120° is different than the others: this result is not explained.
Thank you for the comment. Results are now explained properly (text marked in red in: Results and Discussion). The discussion is now linked to the results. However, the dependency between flexural strength and the dominant grain size in this particular case can’t be clearly explained due to the fact that the matrix is polyphasic and the shells are formed around the grains, which allows for a large margin of error.
Line 220 – 224: the authors asses than « only the materials fired at 1160 and 1180° are suitable for external use »: but which are the results that allow to conclude that, and on which norm or known results it relies?
Thank you for this observation. 10545 standard has been added as [18] and [20] in References. Moreover ASTM C373-18:2018 has been addend as [19] in references and it refers to the boil method for extruded ceramic tiles. The criteria for the final product are described in the Introduction. However, the assumption that the tiles would meet the requirements for external use was too much of a generalization. This assumption is removed.
On figure 3, errors bars are represented for some results but not for all the values (or maybe they’re so small that they can’t be seen?) and we haven’t any information about the number of tests carried on to calculate these error bars.
Error bars are represented for all results. In some cases they are small. The measurement was made at four different samples in order to determine the error. This information is supplemented in the text.
For the colour measurements: how did the authors checked the homogeneity of the samples? The photographs show some differences.
The measurement was made at three different locations on the sample. This information is supplemented in the text.
We would like to thank You for the useful tips that allowed us to improve our manuscript and we hope that our responses to the comments are transparent and sufficient.
We are looking forward to Your response.
Best regards,
Kornelia Wiśniewska
AGH University of Science and Technology, Kraków, Poland

Round 2
Reviewer 1 Report
1 The conclusions include too many detail results. You should outline the main conclusions according to your subject.
2 English language should be checked carefully. For example, at line 167, "The tested masses and sintered samples were prepared according to in laboratory contidions." ; At line 171 ,"The masses prepared the described way were stored in humid conditions for 24 hours."
3 There are still some errors in the manuscript. For example, at line 220, "According to Figure 3a and Figure 3b, a decrease in open porosity ", It should be 2a and 2b.
Author Response
Dear Reviewer,
Thank You for the review. We are submitting our revised manuscript through the Author Center.
Below We are sending responses to Your comments.
1 The conclusions include too many detail results. You should outline the main conclusions according to your subject.
According to Your suggestion we have shortened the conclusions and included only information related to the topic of the article. The remaining conclusions were moved to the discussion.
2 English language should be checked carefully. For example, at line 167, "The tested masses and sintered samples were prepared according to in laboratory contidions." ; At line 171 ,"The masses prepared the described way were stored in humid conditions for 24 hours."
Thank You for noticing this language errors. These errors have been corrected
3 There are still some errors in the manuscript. For example, at line 220, "According to Figure 3a and Figure 3b, a decrease in open porosity ", It should be 2a and 2b.
These errors have also been corrected.
We hope that the improvements we have made to the manuscript and to the attached figures are clear and sufficient.
We are looking forward to Your response.
Best regards,
Kornelia Wiśniewska
AGH University of Science and Technology, Kraków, Poland

Reviewer 2 Report
The paper is now much stronger and significantly improved. As such, I find it suitable to be published.
Author Response
Dear Reviewer,
Thank You sincerely for the recommendation for publication.
Best regards,
Kornelia Wiśniewska
AGH University of Science and Technology, Kraków, Poland

Reviewer 3 Report
You have precised that you've performed color measurement on raw clays. But you should have given some information about how you prepared the sample to make it : I mean that, when pulverulent materials are measured, state of surface and sample compaction are important parameters and you should give information about these parameters.
About the color measurements on the ceramic materials, you mentionned that you measured three points but you've not indicated the dispersion of the results on figure 3.
It remains many typing mistakes, particularly in the added paragraphs
Author Response
Dear Reviewer,
Thank You for the review. We are submitting our revised manuscript through the Author Center.
Below We are sending responses to Your comments.
You have precised that you've performed color measurement on raw clays. But you should have given some information about how you prepared the sample to make it : I mean that, when pulverulent materials are measured, state of surface and sample compaction are important parameters and you should give information about these parameters.
Thank You for this comment. The manuscript has been supplemented with this information.
“In order to determine the color of the described raw material, 10 g of the raw clay was pressed by a hand press to form a 0.5 cm high sample”.
About the color measurements on the ceramic materials, you mentionned that you measured three points but you've not indicated the dispersion of the results on figure 3.
Error bars are represented for all results. The reason why they are not visible is because in the case of color measurement the dispersion of the results is small.
It remains many typing mistakes, particularly in the added paragraphs
Thank You for noticing these errors. We have once again checked the manuscript for linguistic and typing mistakes and corrected all of them.
We hope that the improvements we have made to the manuscript are clear and sufficient.
We are looking forward to Your response.
Best regards,
Kornelia Wiśniewska
AGH University of Science and Technology, Kraków, Poland
